# Protein Changes in Response to Lead Stress of Lead-Tolerant and Lead-Sensitive Industrial Hemp Using SWATH Technology

**DOI:** 10.3390/genes10050396

**Published:** 2019-05-22

**Authors:** Cheng Xia, Li Hong, Yang Yang, Xu Yanping, Huang Xing, Deng Gang

**Affiliations:** 1School of Agriculture, Yunnan University, Kunming, Yunnan 650504, China; chenxia3362623@163.com (C.X.); lhyndx@hotmail.com (L.H.); wsyjy198797@163.com (Y.Y.); denggang1986@ynu.edu.cn (D.G.); 2College of Agriculture and Life Sciences, Kunming University, Kunming, Yunnan 650241, China; chenxia3362623@163.com; 3Industrial Crop Research Institute, Yunnan Academy of Agricultural Sciences, Kunming, Yunnan 650205, China; jingzuosuo@163.com; 4Environment and Plant Protection Institute, Chinese Academy of Tropical Agricultural Sciences, Haikou, Hainan 571101, China; hxalong@gmail.com

**Keywords:** Pb stress, Industrial hemp (*Cannabis sativa* L.), SWATH, Pb-stress adaption

## Abstract

Hemp is a Pb-tolerant and Pb-accumulating plant and the study of its tolerance mechanisms could facilitate the breeding of hemp with enhanced Pb tolerance and accumulation. In the present study, we took advantage of sequential window acquisition of all theoretical mass spectra (SWATH) technology to study the difference in proteomics between the leaves of Pb-tolerant seed-type hemp variety Bamahuoma (BM) and the Pb-sensitive fiber-type hemp variety Yunma 1 (Y1) under Pb stress (3 g/kg soil). A total of 63 and 372 proteins differentially expressed under Pb stress relative to control conditions were identified with liquid chromatography electro spray ionization tandem mass spectrometry in BM and Y1, respectively; with each of these proteins being classified into 14 categories. Hemp adapted to Pb stress by: accelerating adenosine triphosphate (ATP) metabolism; enhancing respiration, light absorption and light energy transfer; promoting assimilation of intercellular nitrogen (N) and carbon (C); eliminating reactive oxygen species; regulating stomatal development and closure; improving exchange of water and CO_2_ in leaves; promoting intercellular transport; preventing aggregation of unfolded proteins; degrading misfolded proteins; and increasing the transmembrane transport of ATP in chloroplasts. Our results provide an important reference protein and gene information for future molecular studies into the resistance and accumulation of Pb in hemp.

## 1. Introduction

The heavy metal lead (Pb) is a significant threat to human health, agricultural production and ecological safety. Soil Pb levels have been steadily increasing as a result of human activities and it is now one of the most serious heavy metal pollutants. Pb pollution can persist for more than 300 years with very slow degradation and it can accumulate in soil and living organisms [1]. After absortion by humans, 90% of Pb remains in the skeleton and it can be transferred to the next generation through pregnancy and lactation [2]. Pb has a substantial negative impact on the nervous system, intellectual development, renal function and physical development of infants and children. Mild Pb load can cause neurophysiological damage [2] but severe Pb toxicity leads to genetic mutations and carcinogenesis, which are irreversible [3]. Therefore, the remediation of Pb pollution in the environment is one of the priorities of current studies.

Pb is not required for normal plant physiology and Pb accumulation is usually toxic. Most plants are susceptible to Pb pollution and only a few can tolerate Pb accumulation, including members of the Brassicaceae, Caryophyllaceae, Poaceae, Violaceae and Fabaceae families [4,5,6] but such Pb-accumulators usually exhibit slow growth and low biomass [7]. How to develop Pb-tolerant plants with a high economic benefit, large biomass and rapid growth is an urgent question to be resolved [8]. One feasible approach is the development of new plant varieties with high Pb-accumulation capacity by identifying genes associated with Pb uptake and tolerance using molecular technology and other biotechnologies. Several proteins have been demonstrated to play roles in Pb uptake and tolerance, including NtCBP4 (*Nicotiana tabacum* calmodulin- binding protein) [9], YCF1 (a cadmium factor in yeast) [10] and TaPCS1 (the low-affinity cation transporter 1 in *Triticum aestivum*) [11]. Further investigations are needed to identify more related proteins and genes associated with Pb uptake, accumulation or tolerance. 

Because of its wide distribution, well-developed root system, rapid growth and high economic value, Industrial hemp (*Cannabis sativa* L.) plays a significant role in the textile, manufacturing and food industries [12]. Hemp shows a strong Pb tolerance and accumulation ability and it is listed as one of the cultivated crops with Pb tolerance, strong adaptability and high biomass. Hemp can normally germinate in the presence of 1 g/L Pb, where its germination rate is >80% [13,14,15,16]. In addition, hemp can grow in soil with a high level of Pb pollution and it can tolerate Pb accumulation in surface tissues to a level of 26.3 kg/hm^2^ [14,15,16]. The mechanism of Pb uptake, accumulation and tolerance in hemp has not been reported and the molecular mechanism of Pb tolerance and accumulation is still not clear. Thus, there is a need to investigate the proteins and metabolic pathways associated with Pb tolerance and accumulation. One strategy could be to use proteomics to study the physiological process of adaptation to Pb stress in hemp.

In this study, we used high-throughput proteomics technology to analyze the proteins and metabolic pathways, particularly photosynthesis, energy metabolism and stress response, related to Pb stress in two hemp varieties (one Pb tolerant and one Pb sensitive). These data will provide important molecular reference information for future investigations into Pb tolerance and accumulation in hemp. 

## 2. Materials and Methods

### 2.1. Cultivation

We used the seed-type hemp variety Bamahuoma (BM) and fiber-type hemp variety Yunma 1 (Y1). While each variety exhibits some Pb tolerance, BM exhibits greater tolerance than Y1, so BM and Y1 were referred to as Pb tolerant and Pb sensitive, respectively. Y1 and BM seeds were kindly provided by the Industrial Corps Institute of Yunnan Academy of Agricultural Sciences and the Guangxi Academy of Agricultural Sciences, respectively.

The pot method was used with a pot height of 24 cm, diameter of 18 cm and holes at the bottom. Each pot was filled with 12 kg bulk material (red soil: humus soil = 1:1 with 17% moisture content) that contained no Pb. Two treatments were performed for each variety: Pb stress, in which Pb(NO_3_)_2_ was applied at 3 g/kg soil dry weight and a control, without Pb treatment. Soil was mixed well with the Pb, put into the pots, watered and allowed to sit for 1 week before planting. Fifteen replicate pots were prepared for each variety/treatment combination and each pot contained six hemp plants. A total of 60 pots were placed in a green house and normal management practices were applied.

### 2.2. Determination of Physiological and Biochemical Indexes

Obvious Pb-toxicity symptoms were observed in both varieties when the average height of the control plants was ~40 cm on average (about 40 days after sowing). Measurements were carried out on the third to fifth leaves from the plant base using a Li-6400 Portable Photosynthetic System (LI-COR, Lincoln, NE, USA) for photosynthetic parameters and spectrophotometry for physiological indicators (soluble protein content, chlorophyll content, malondialdehyde content (MDA) and superoxide dismutase (SOD)) [17]. In addition, 2–4 g of the third to fifth fully expanded leaves from the bottom was randomly collected from three different hemp plants in three different pots undergoing each of the samples (2 treatments × 2 specimens), with three biological replicates. The 12 samples were immediately snap-frozen and stored at −80 °C prior to protein extraction.

### 2.3. Protein Preparation

Each of the 12 samples was ground to powder in liquid nitrogen with a mortar and pestle, then dissolved with the lysis buffer (7 M urea, 2 M thiourea, 4% (w/v) CHAPS, 40 mM Tris-HCl, 1 mM phenylmethylsulfonyl fluoride, 10 mM dithiothreitol) by sonication for 10 min and centrifuged at 25,000× *g* for 15 min at 4 °C. The supernatant was precipitated with four volumes of prechilled 10% trichloroacetic acid (TCA)-acetone at −20 °C overnight and centrifuged at 25,000× *g* for 30 min at 4 °C. The precipitate was washed with prechilled acetone three times. The protein was dissolved with the lysis buffer and protein concentration was determined by the Bradford assay. The samples were reduced with 10 mM DTT for 1 h at 56 °C and alkylated with 55 mM iodoacetamide for 45 min at room temperature in darkness. 100 μg proteins for each sample were digested by the filter-aided sample preparation method with trypsin (Promega, Madison, WI, USA), with 0.1 mM triethylammonium bicarbonate as the buffer solution [18]. Ten μg peptides each sample were pooled and vacuum-dried for building the library.

### 2.4. Fractionation by High pH Reverse Phase Chromatography

For high pH reverse phase chromatography using the LC-20AB HPLC Pump system (Shimadzu, Kyoto, Japan), the pooled peptides were reconstituted with 0.1 mL buffer A (20 mM ammonium formate, 2% acetonitrile (ACN), pH 10) and loaded onto a Gemini-NX 5 μ c18 110A 150 × 4.6 mm column (Phenomenex, Guangzhou, China). The peptides were eluted at a flow rate of 1 mL/min with a gradient of buffer A for 10 min, 5–30% buffer B (20 mM ammonium formate (FA), 98%ACN, pH 10) for 20 min and 30–80% buffer B for 2 min. Elution was monitored by measuring absorbance at 214 nm and fractions were collected every 1 min. The eluted peptides were pooled as 6 fractions and vacuum-dried.

### 2.5. LC-ESI-MS/MS Analysis

Each fraction was resuspended in buffer A (2% ACN, 0.1%FA) and centrifuged at 20,000× *g* for 10 min and 2 μg of peptide was loaded onto nanoLC 425 system (Eksigent part of SCIEX Dublin, CA, USA) with an analytical C18 column (inner diameter 75 μm) packed in-house. The gradient (with a flow rate of 300 nL/min) was set up as follows: 5–20% solvent B for 45 min, 20–24% solvent B for 10 min; 24–30% solvent B for 15 min; 30–35% solvent B for 15 min; 35–90% solvent B for 15 min. Data-dependent acquisition (DDA) was performed with a TripleTOF 5600 System (SCIEX, Concord, ON, Canada). The source parameters were set up as follows: ion spray voltage at 2.3 kV, curtain gas at 30 PSI, ion source gas at 20 PSI and an interface heater temperature of 150 °C. Mass spectrometry (MS) was performed in a high-resolution mode (>30,000 fwhm) for TOF (Time of Flight) MS scans. For the DDA model, the MS range was 350 to 1250 Da with ion accumulation time of 250 ms. Tandem MASS spectrometry (MS/MS) data were acquired in 50 ms and as many as 40 product ion scans were collected if they exceeded a threshold of 125 counts per second (counts/s) and with a 2^+^ to 4^+^ charge-state. Dynamic exclusion was set for 1/2 of peak width (15 s). For the sequential window acquisition of all theoretical mass spectra (SWATH) model, the MS range was 350 to 1250 Da with an iron accumulation time of 50 ms, with 57 fixed windows used at 50 ms per SWATH scan with high sensitivity mode with resolution 15 K for *m/z* 100–1250. All MS data is uploaded to the iProX (https://www.iprox.org, IPX0001592000).

### 2.6. Library Generation and SWATH Data File Process

Data generated by DDA were searched by Protein Pilot 5.0 (AB SCIEX) with the Paragon algorithm against the hemp transcriptome database (http://genome.ccbr.utoronto.ca/downloads.html, finola1_transcriptome-full.fa.gz). A false discovery rate (FDR) analysis was performed. The output group file was used as the reference spectral library. Then SWATH analysis was processed with the SWATH Acquisition MicroApp 2.0 in PeakView^®^ Software (SCIEX, Foster City, CA, USA) with the following parameters: five peptides/protein, five transitions/peptide, peptide confidence level of > 90%, FDR < 1%, excluded shared peptides and Ion Library Mass Tolerance of 100 ppm. The quantitation data was normalized with the median. Three biological replicates for each sample were performed for t-test analysis. The protein levels with a *p*-value less than 0.05 and ratio ≥1.5 (increase) or ≤0.67 (decrease) were considered significantly different in our experiment.

## 3. Results and Discussion

### 3.1. Physiological Indicators under Pb Stress

The growth stage at which the plants were sampled was selected to coincide with rapid growth (40d after sowing). The indicators of Pb stress in BM were more stable than those in Y1, especially the net photosynthetic rate and MDA level (Table 1), indicating that the Pb tolerance of BM was higher than that of Y1 during rapid growth, which can also be verified in the hemp growth (Figure 1).

Pb stress affected all the photosynthetic indicators in hemp. The net photosynthetic rate (Photo), stomatal conductance (Cond) and transpiration rate (Tromol) of Y1 under Pb stress were reduced by 19%, 12.8% and 14.4%, respectively, compared to the controls and the decrease in net photosynthetic rate was significant. However, the intracellular CO_2_ (Ci) concentration was increased by 5% under Pb stress. In BM, only the net photosynthetic rate was decreased by 4.7% due to Pb stress and all other indicators were higher than those in the controls. Among these, the Ci concentration was significantly elevated by 10.2%. Thus, Pb-tolerant BM exhibited a reduction (albeit much less than in Y1) in photosynthetic rate upon Pb treatment but it could adapt to the stress by increasing the Ci, Cond and Tromol values.

The effects of Pb stress on soluble protein content, chlorophyll, SOD and MDA showed that the first three indicators decreased under Pb stress in both BM and Y1 but the MDA content of Y1 was significantly elevated by 62.3% under Pb stress, whereas only a 4% non-significant increase was observed in BM. MDA is one of the products of membrane lipid peroxidation, so higher MDA suggests greater damage to the membranes. Therefore, it was revealed that Pb stress resulted in greater damage to the membrane system of Y1 than that of BM; that is, the membrane system of Y1 was more susceptible to Pb stress.

### 3.2. Protein Identification and Analysis

A total of 2131 proteins were identified in the SwissProt/UniProt database (Appendix A) and 63 and 372 were recognized as being differentially expressed (≥1.5-fold) under Pb stress compared to the control conditions in BM (Appendix A) and Y1 (Appendix A), respectively. Of these, 39 and 231 proteins in BM and Y1, respectively, were upregulated under Pb stress, which accounted for 61.9% and 62.1% of the differentially expressed proteins in BM and Y1, respectively, suggesting that the metabolic status and molecular physiological activities of the Pb-tolerant variety BM were more stable than those of the Pb-sensitive variety Y1 under Pb stress and the high metabolic activity in Y1 resulted in Pb sensitivity.

The different proteins in each cultivar were functionally classified according to the method proposed by Bevan et al. (1998) [19] (Figure 2 and Figure 3). They were classified into 14 categories: primary metabolism, energy, protein destination and storage, disease/defense, protein synthesis, photosynthesis, transport, transcription, signal transduction, cell structure, secondary metabolism, intracellular traffic, cell growth/division and unknown. Over 67% of proteins associated with protein destination, storage and transcription were downregulated in the differentially expressed proteins of Y1, whereas >91% of proteins associated with transporters and photosynthesis in this variety were upregulated. In BM, proteins associated with transporters, protein synthesis, energy and secondary metabolism were upregulated under Pb stress, while proteins related to cell growth/division, intracellular traffic and photosynthesis were downregulated (Figure 2 and Figure 3).

### 3.3. Proteins Related to Energy Metabolism

Under Pb stress, respiration is increased to provide more energy and stored biomass is consumed to produce more adenosine triphosphate (ATP) for adapting to the stress. The ATP/ADP ratio in pea and barley is increased and their respiratory rates are elevated by >50% under Pb stress [20,21]. We showed that many proteins associated with ATP biosynthesis were upregulated in Y1 under Pb stress, namely ATP synthase subunit a (P56758, P56757), ATP synthase protein MI25 (Q04613), ATP synthase protein YMF19 (P93303), nucleoside diphosphate kinase III (O49203), pyruvate kinase (PKE) (Q94KE3, Q9FNN1, Q9FM97) and adenylate kinase 5 (ADK) (Q8VYL1). PKE is the key enzyme in the final step of glycolysis that produces pyruvic acid and ATP. The resulting pyruvic acid then enters the TCA cycle to increase respiration. ADK regulates the ATP content as well as the level of other adenyl phosphates (for example: ADP and AMP) in the cells [22]. In addition, two proteins related to the respiratory electron chain, NADH dehydrogenase (ubiquinone) iron–sulfur protein (Q95748, Q42577), were upregulated in Y1. This allows efficient transportation of electrons from NADH to the respiratory chain to enhance respiration [23].

Other proteins associated with energy were downregulated in Pb-stressed Y1, namely *β*-1,3-glucanase-like protein (Q9M2T6) for catalyzing polysaccharide degradation; apyrase 1 (APE) (Q9SQG2) for catalyzing hydrolysis of anhydride bonds in nucleoside triphosphate and nucleoside diphosphate; the key enzyme in the sugar degradation pathway, fructose bisphosphate aldolase 6(Q9SJQ9), probablyfructokinase-4 (FRK) (Q9M1B9) for the decomposition of sucrose and fructose in glycometabolism and the respiratory electron transporters cytochrome C oxidase subunit 6b-1 (Q9S7L9) and cytochrome C oxidase subunit 5b-1 (Q9LW15). APE regulates ATP levels and mediates stomatal closure [24,25]. FRK affects microtubule development and the transportation of growth-related carbohydrates, water and minerals [26] and it is also upregulated under other stresses, such as salt [27], drought [28] and anoxia [29].

Two energy-related proteins, aminotransferase 3 (P46644) and aldose 1-epimerase (Q9LVH6) were upregulated in BM. The former is the key enzyme in the amino acid cycle and Calvin cycle, which plays roles in C and N metabolism and energy metabolism to strengthen the connection between photosynthesis and respiration [26]. The latter is the key enzyme in glycolysis [30].

Pb stress induced ATP synthesis in hemp to improve energy metabolism and enhance respiration and its association with photosynthesis, leading to stress adaptation. The inhibition of carbohydrate catabolism and electron transfer of respiration were observed in the Pb-sensitive Y1.

### 3.4. Proteins Related to Photosynthesis

Photosynthesis disorders are one of the first symptoms of abiotic stresses [31,32] and heavy metals have a serious impact on plant photosynthesis, resulting in damage of the chloroplast structure, reduction of chlorophyll content, stomatal closure and interference with the electron transfer chain [33,34]. Under Pb stress, the net photosynthetic rate and chlorophyll content decreased in both varieties of hemp (Table 1). Many photosynthetic proteins (87%) were upregulated in Pb-stressed Y1, whereas no differentially expressed photosynthetic proteins were identified in the Pb-tolerant BM. This indicates that photosynthesis of Pb-tolerant hemp is stable even under Pb stress and that Pb-sensitive hemp has the potential to increase photosynthesis, which is consistent with the photosynthetic indicators in BM and Y1. The upregulated photosynthetic proteins in Pb-stressed Y1 mainly consisted of photosystem I-related proteins, such as photosystem I reaction center subunit V (Q9S7N7), Photosystem I reaction center subunit XI (Q9SUI4) and so forth, and photosystem II-related proteins (P56777, Q9SHE8, etc.), suggesting that Y1 enhances both photosystems to increase light absorption and transfer [35,36]. In addition, the chlorophyll biosynthesis-related protein, chlorophyll a-b binding protein (e.g., Q9XF88, Q9SHR7, etc.), was also upregulated in Y1, as well as the activity of BC1 complex kinase 1 (Q8RWG1), which prevents chloroplast light damage and the degradation of the photosystem II core. These data indicate that chlorophyll biosynthesis was increased and damage to the photosynthetic system was avoided in Y1 under Pb stress [37,38]. Other than photosynthesis, proteins associated with photorespiration, glycerate dehydrogenase HPR (Q84VW9) and ferredoxin-dependent glutamate synthase 1 (Q9ZNZ7, FDGS), were also upregulated in Y1, suggesting that photorespiration is also enhanced in Y1. FDGS also plays a role in the assimilation and reuse of ammonium ions in photorespiration [39]. Due to the small effect of Pb on photorespiration, enhancement of photorespiration affects C assimilation in plants [40].

Only two photosynthetic proteins, pyruvate dehydrogenase E1 component subunit *β*-3 (O64688) and ferredoxin-3 (Q9ZQG8) were downregulated in Pb-stressed Y1. The former converts pyruvate to acetyl-CoA and CO_2_ in the Kelvin cycle [41], while the latter facilitates the excitation of chlorophyll by light to accept electrons and electron transfer to ferredoxin: NADP^+^ oxidoreductase.

Photosynthesis and photorespiration were simultaneously strengthened in Y1 under Pb stress, showing potential to increase photosynthesis but the process of chlorophyll electron transfer is inhibited.

### 3.5. Primary and Secondary Metabolism-Related Proteins

Under Pb stress, primary and secondary metabolism of hemp is affected and plants produce metabolites associated with Pb tolerance in order to adapt to the stress. Upregulated primary-metabolism-related proteins accounted for 69% and 57% of the total upregulated proteins in Y1 and BM, respectively, while the proportion was 64% and 100% for the secondary-metabolism-related proteins.

The abundance of some primary metabolites increased in response to Pb stress in Y1, namely proteins related to starch metabolism (hosphoglucan phosphatase DSP4 (Q9FEB5)), proteins associated with amino acid metabolism (homocysteine S-methyltransferase 1 (Q9SDL7) and asparagine synthetase (glutamine-hydrolyzing) 1 (P49078, APSE)), proteins related to carbohydrate metabolism (*β*-galactosidase 1 (A7WM73, GTE)) and proteins related to other primary metabolism (pyridoxal phosphate homeostasis protein (F4JVS4, PPHP), alkaline/neutral invertase E (Q9FK88) and nitrate reductase (NADH) 2 (P11035, NRD)). The carbohydrate-metabolism-related protein APSE enhances the assimilation, allocation and reactivation of N through the phloem [42]. GTE degrades galactose in the cell wall, thus making the cell wall loose to facilitate cell expansion [43]. PPHP regulates intracellular stability by synthesizing pyridoxal 5′-phosphate [44]. Alkaline/neutral invertase E participates in the C flow between the cytoplasm and plasmids of the leaves, development of the photosynthetic apparatus and N assimilation during the seedling stage [45]. NRD plays roles in N absorption and assimilation in plants [46].

The upregulated proteins in Pb-stressed BM were mainly associated with aspartate catabolism [isoaspartyl peptidase/*L*-asparaginase 1andextensins (P50287)] and serine metabolism (peptidyl serine *α*-galactosyltransferase (Q8VYF9)). We showed that acid phosphatase, which is related to phosphate absorption, was downregulated in Y1. Pb stress always leads to P deficiency because phosphate precipitation is triggered by Pb and the downregulation of acid phosphatase, further indicating P deficiency in Y1 under Pb stress [47]. Therefore, metabolism in Y1 and BM was enhanced under Pb stress, with Y1 exhibiting improvement of N metabolism, increased intercellular stability and C flow but P deficiency may also have resulted from Pb stress.

Secondary metabolism proteins were all upregulated in BM but only 69% were upregulated in Y1. The upregulated secondary metabolism proteins in both varieties were all related to synthesis of photosynthetic pigments and chloroplast protection, such as protein lutein deficient 5 (Q93VK5), lycopene epsilon cyclase (Q38932), zeta-carotene desaturase (Q38893) and probable carotenoid cleavage dioxygenase 4 (O49675) of Y1 and ferrochelatase-2 (O04921) and protein lutein deficient 5 (Q93VK5) of BM. However, GDSL esterase/lipase (Q9C7N4, Q9LU14), isopentenyl-diphosphate delta-isomerase I (Q38929, IDPD) and membrane steroid-binding protein 2 (Q9M2Z4) were downregulated in Pb-stressed Y1, suggesting that the biosynthesis of chlorophyll in Y1 may be suppressed by reduction of IDPD activity [48]. In addition, the downregulation of GDSL esterase/lipase decreases lipid synthesis, along with membrane steroid-binding protein 2, so it is believed that the cell membrane structural damage in Y1 caused by Pb stress lowers the stress resistance.

### 3.6. Protein Related to Disease/Defense

Pb stress can induce a high level of reactive oxygen species (ROS), such as superoxide anion radicals, hydroxyl radicals, H_2_O_2_ and alkoxy radicals, which react with cell membranes, organelles, proteins, lipids and nucleotides to cause abnormal cell metabolism and even cell death [49,50]. Plant tolerance of Pb is often presented in terms of increased antioxidant capacity and we found that many antioxidant proteins were upregulated in Pb-stressed Y1 and BM, such as SOUL heme-binding family protein (F4K452), glutathione reductase (P48641, P42770, GHR), linoleate 9S-lipoxygenase 1 (Q06327) and lipoxygenase (P38418, Q9FNX8, LPX) and their upregulation facilitated scavenging of accumulated peroxide. Linoleate 9S-lipoxygenase 1 was differentially expressed in both varieties under Pb stress compared to non-stress conditions, with up- and downregulation in BM andY1, respectively. This protein is also involved in the growth and development of plants, in delaying senescence and in regulation of lateral root growth [51]. LPX plays an importation role in the jasmonate pathway of induced disease resistance and in the prevention of lipid peroxidation to form ethylene, which may lead to leaf senescence [52]. GHR mainly prevents the conversion of oxidized glutathione to glutathione, leading to a reduction of H_2_O_2_ [53]. Both proteins were upregulated in Pb-stressed BM and Y1.

Probable carboxylesterase 8 (O64640), *β*-carbonic anhydrase 1 (P27140) and Probable carboxylesterase 8 (O64640), carbonic anhydrase 2 (P42737) were upregulated in Pb-stressed Y1 and BM, respectively. These proteins are associated with H_2_CO_3_ metabolism and catalyze conversion of CO_2_ and water to bicarbonate and water, transmitting CO_2_ signals to regulate stomatal development and movement, which, in turn, mediates gas exchange between plants and the atmosphere [54,55]. The upregulation of these proteins is consistent with the increase of Ci in BM and Y1 under Pb stress (Table 1).

Pb can induce stomatal closure, resulting in deficiency of CO_2_. Two stomatal-closure-regulating proteins were identified as being differentially expressed in Pb-stressed Y1: bifunctional D-cysteine desulfhydrase/1-aminocyclopropane-1-carboxylate deaminase (F4HYF3) and phospholipase D α1 (Q38882), the upregulation of which facilitates stomatal closure to allow CO_2_ to enter the leaves [56,57].

Under Pb stress, hemp is vulnerable to abiotic stresses due to damage to the cell wall and membrane. Proteins related to fungal pathogen and pest resistance were upregulated in Y1, including acidic endochitinase, basic endochitinase B (P19171), cinnamyl alcohol dehydrogenase 8 (Q02972), patatin-like protein 2 (O48723) and tetratricopeptide repeat (TPR)-like superfamily protein (Q94K88). The first three are involved in the repair of the cell membrane structure [58].

Protein plant cadmium resistance 2 (Q9LQU4), a protein associated with heavy metal absorption, was upregulated in Y1. This protein is involved in the excretion and long distance transport of zinc and cadmium in plants and it can transport toxic substances into the xylem to detoxify the epidermal cells [59], so this protein may play a role in the transportation and detoxification of Pb.

### 3.7. Proteins Related to Protein Synthesis and Transcription

The differentially expressed proteins associated with protein synthesis were all upregulated in BM, compared with only 61% in Y1, indicating that protein synthesis plays a vital role in Pb tolerance. Of these, 34 were annotated as ribosomal proteins, accounting for 71% of the total differentially expressed proteins related to protein synthesis. Ribosomal proteins are mainly involved in the biosynthesis of ribosome proteins, plant metabolism and multiple stress resistance [60].These proteins were associated with tolerance to Pb stress, especially in BM and such an effect has been previously reported in hemp under salt stress [61].The abundance of proteins related to translation, such as lysine-tRNA ligase (Q9ZPI1), serine-tRNA ligase (Q39230) and eukaryotic translation initiation factor 3 subunit A (Q9LD55) and subunit F (O04202) was substantially reduced in Y1 but the T10F20.8 protein (Q9LMS7) was upregulated in BM. This indicated that protein synthesis in Y1 was inhibited by Pb stress, affecting cell proliferation and cell cycle progression, which might have been responsible for the weaker Pb tolerance exhibited by Y1.

The transcription proteins of Y1 exhibited a similar expression pattern to translation proteins and downregulated proteins contributed to 75% of the differentially expressed proteins in this category but only two transcription-related proteins were reduced in BM. However, two proteins, transcription initiation factor TFIID subunits 9 (Q9SYH2) and TFIID subunit 7 (B9DG24) were upregulated in Pb-stressed plants of both BM and Y1, with these transcription factors playing indispensable roles in genetic regulation and tolerance to various abiotic stresses in plants [62]. Thus, the increase in abundance of these two proteins ensures the transcription process in Pb-stressed hemp.

### 3.8. Proteins Related to Transport and Signal Transduction

Proteins related to transport were all upregulated in Pb-stressed BM, while only one of 22 transport proteins in Y1 was downregulated, indicating that transport proteins are important for Pb tolerance in hemp. The upregulated proteins in Y1 included five water transport-related aquaporins (e.g., Q06611, P25818, etc.), patellin (Q56ZI2, Q56Z59) for transmembrane transport of hydrophobic materials and proteins associated with transportation of organic compounds, mitochondrial dicarboxylate/tricarboxylate transporter DTC (Q9C5M0), mitochondrial phosphate carrier protein 3 (Q9FMU6), mitochondrial carnitine/acylcarnitine carrier-like protein (Q93XM7), MD-2-related lipid recognition domain-containing protein/ML domain-containing protein (F4J7G5) and ras-related protein RABA2a (O04486), all of which are responsible for the intracellular transport of organic compounds and water to ensure various life processes, including respiration, photosynthesis, photorespiration, N and C assimilation and growth hormone biosynthesis [63,64,65,66,67]. The abundance of the ABC transporter E family member 2 (Q8LPJ4) was increased in Pb-stressed Y1 and it has been shown to participate in the transport of cytotoxic substances, including heavy metals Pb and Cd, which facilitates the detoxification of heavy metals [68,69]. Nuclear pore complex protein NUP93A (O22224) is responsible for the transport of RNA and ribosomal proteins from the nucleus to the cytoplasm and the transport of signal molecules and lipids into the nucleus. The abundance of this protein was elevated in Pb-stressed BM, which is consistent with upregulation of ribosomal proteins in BM.

The abundance of all three differentially expressed proteins related to signal transduction were reduced in Pb-stressed BM, compared with that of two of the three proteins in Y1, suggesting that the signal transduction pathways were compromised in hemp under Pb stress. All the proteins associated with Ca^2+^ signal transduction were downregulated in BM and Y1, such as Probable calcium-binding protein CML13 (Q94AZ4, in Y1), CML27 (Q9LE22, in BM) and calreticulin (e.g., O04151, in Y1), indicating that Pb stress inhibits signal transmission in cell membranes [70,71]. Abundance of a calcium-sensing receptor (Q9FN48), which is related to stomatal closure and leaf abscission [72], was increased in Y1.That is, Y1 adapted to Pb stress by regulating stomatal closure and leaf abscission via this protein and stomatal conductance was reduced by 12.8% in Y1 compared to the controls (Table 1). Receptor-like protein kinase FERONIA (Q9SCZ4) was downregulated in Y1, which may lead to shortened vegetative growth stage of cells and suppressed root hair growth [73,74].

### 3.9. Proteins Related to Protein Destination and Storage

Over 66% of the differentially expressed proteins related to protein destination and storage were upregulated in Pb-stressed BM, compared with only 33% in Y1. Abundance of the T-complex protein 1 subunit (Q84WV1, P28769) and GrpE protein homolog (Q9XQC7, Q8LB47) increased in BM and decreased in Y1 under Pb stress, indicating that both proteins play important roles in Pb tolerance. These two proteins, as molecular chaperones, protect unfolded proteins from polymerization and degrade misfolded proteins [45,75]. The abundance of other folding-related proteins was also elevated in Y1, namely *α*/*β*-hydrolases superfamily protein (F4K2Y3), chaperone protein dnaJ A6 (Q9SJZ7), ATP-dependent Clp protease proteolytic subunit 6 (Q9SAA2) and peptidyl-prolylcis-trans isomerase FKBP16-4 (Q9SR70). The abundance of some proteins related to folding was reduced in Pb-stressed Y1, such as chaperonin (e.g., P29197) and 26S proteasome non-ATPase regulatory subunit (e.g., Q9LNU4). The abundance of two photosynthesis-related proteins—protease Do-like 2 (O82261) and serine protease SPPA (Q9C9C0)—was upregulated in Pb-stressed Y1. Both proteins are involved in the protection of Photosystem II and degradation of light-damaged proteins and both exhibited strong light protective effects [76,77], which is consistent with the upregulation of photosynthetic proteins in photosystem II of Y1. The abundance of subtilisin-like protease (Q9LVJ1, O65351) was increased in Pb-stressed BM and this protease is associated with the plant cell cycle to maintain normal cell growth [78]. The abundance of three ubiquitin-related proteins was downregulated in Y1 (e.g., Q9SII9). Proteins tagged by ubiquitin are identified and degraded rapidly and they usually participate in DNA repair and cell-cycle regulation [79], which also might be one of the reasons for the weaker growth of Y1 compared to BM under Pb stress (Figure 1).

### 3.10. Proteins Related to Cell Growth/Division, Cell Structure and Intracellular Traffic

Only one cell growth/division protein was differentially expressed in Pb-stressed BM but 16 were identified in Y1, with half of them downregulated. FRIGIDA-like protein 4b (Q940H8) was increased in Y1 but reduced in BM. It is involved in flowering regulation [80] and flowering time control protein FY (Q6NLV4) was also upregulated in Y1. These data suggest that the reproductive stage of Y1 is fulfilled ahead of schedule by regulating the time of flowering to avoid Pb stress and this is consistent with downregulation of the signal transduction protein receptor-like protein kinase FERONIA (Q9SCZ4) in Pb-stressed Y1. The decrease in abundance of two root-growth-related proteins, fasciclin-like arabinogalactanprotein (e.g., O22126) [81] and leucine-rich repeat extensin-like protein 2 (O48809) [82] and one protein related to the growth of plant root tips and stem apex meristem, endoplasmin homolog (Q9STX5) [82], indicates that the growth of plant roots and stem tips is inhibited, which may be another reason for the compromised plant height of Y1under Pb stress (Figure 1).

Pb inhibits the synthesis of chlorophyll through chlorophyll enzyme activity [83] and Protein activity of BC1 complex kinase 1 (Q8RWG1) [38] associated with chlorophyll degradation was upregulated in Y1. Such upregulation, along with the decrease of membrane steroid-binding protein 2 (Q9M2Z4), could have reduced the chlorophyll content in Y1 under Pb stress (Table 1), although the abundance of chlorophyll a-b binding protein (e.g., Q9XF88),—a chlorophyll biosynthesis-related protein—was elevated. Pb has comparable physical size and oxidation state to other nutrient elements, therefore, it can enter cells by competing for ion channels, resulting in damage of the cell membrane [84]. Therefore, a protein associated with chloroplast membrane structure—thylakoid luminal 15 kDa protein (O22160) [85]—was downregulated, as were the tubulin proteins (Q9ASR0) [86] and bifunctional dTDP-4-dehydrorhamnose 3,5-epimerase/dTDP-4-dehydrorhamnose reductase (Q9LQ04) [87], which is related to cell wall structure.

Both differentially expressed proteins associated with intracellular traffic in BM were downregulated, yet most of the corresponding proteins in Y1 were upregulated. The abundance of proteins associated with transmembrane carrier protein and chloroplast transmembrane transport—such as protein TIC 21(Q9SHU7)—outer envelope membrane protein 7 (Q9SVC4) and outer envelope pore protein 24A (Q1H5C9) was elevated in Y1, which is consistent with upregulation of proteins related to ATP synthesis and photosynthesis in Y1. The abundance of other transmembrane transport proteins, such as ATPase 5, plasma membrane-type (Q9SJB3), transmembrane 9 superfamily member 1 (Q940G0) and transmembrane protein, putative (Q94F10) were also increased in Pb-stressed Y1. However, downregulation of peroxisomal membrane protein PEX14 (Q9FXT6) in Y1 suggests that the transmembrane transport of peroxisomes is inhibited [88], which may lower the capability of antioxidant and anti-aging activities in Y1. In Pb-stressed BM, the mitochondrial import receptor subunit TOM20-2 (P82873) was downregulated. This protein is responsible for the identification and transport of precursors of mitochondrial synthesis and promotion of proteins entering the mitochondrial outer membrane [89].

## 4. Conclusions

We used high-throughput SWATH technology to investigate the tolerance mechanisms of Pb-tolerant BM and Pb-sensitive Y1 under Pb stress. There were 63 and 371 differentially expressed proteins in Pb-stressed BM and Y1, respectively, which were classified into 14 categories. The following responsive mechanisms are used by hemp to cope with Pb stress: (1) Increasing ATP biosynthesis; (2) enhancing respiration; (3) strengthening the connection between respiration and photosynthesis; (4) improving photosynthesis; (5) promoting N and C assimilation; (6) eliminating reactive oxygen species; (7) regulating stomatal development and closure to increase air exchange; (8) elevating water transportation; (9) shortening the growth period and controlling flowering time; (10) protecting unfolded proteins from aggregation and degrading misfolded proteins; and (11) boosting the transmembrane transport of ATP and within chloroplasts. Our results provide important reference information on the proteins involved in Pb tolerance in hemp for future studies.

In this study, we used SWATH technology to study the difference in proteomics between the leaves of Pb-tolerant seed-type hemp variety BM and the Pb-sensitive fiber-type hemp variety Y1 under Pb stress. The response mechanism of BM to Pb stress and the important proteins related to the response mechanism were obtained. However, the important proteins obtained in this study have not undergone subsequent functional verification. Therefore in subsequent studies, it is still necessary to verify the function of these proteins by using the gene cloning technology in the future research.

## Figures and Tables

**Figure 1 genes-10-00396-f001:**
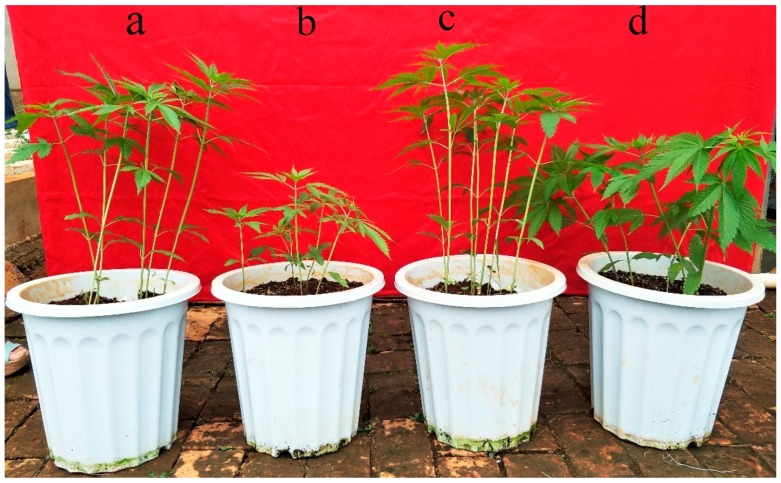
Comparison of hemp growth during rapid growth stage of two varieties under Pb stress. (**a**) Yunma 1 (Y1) control, (**b**) Y1 Pb treatment, (**c**) Bamahuoma (BM) control and (**d**) BM Pb treatment. Soil with 3 g/kg Pb and without Pb were referred to as Pb stress and control, respectively.

**Figure 2 genes-10-00396-f002:**
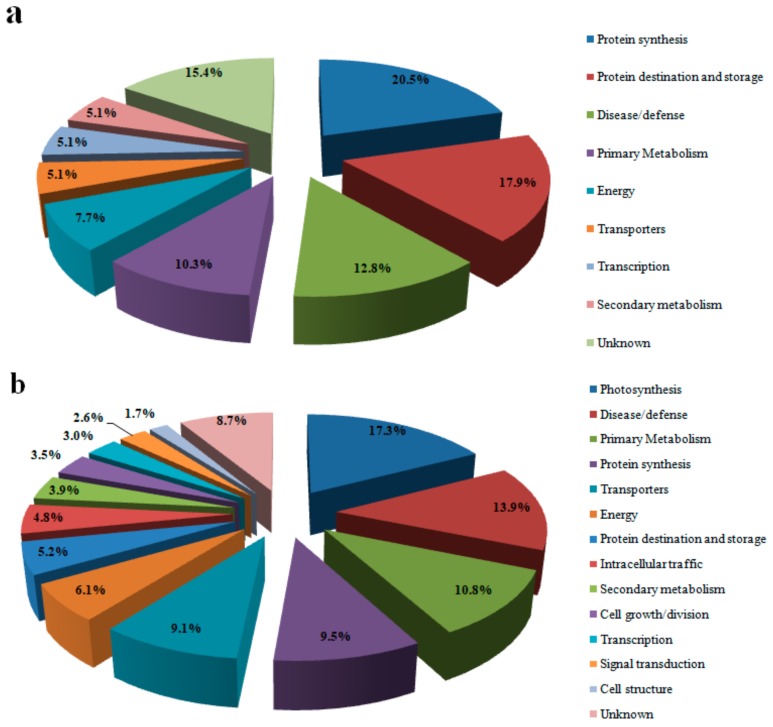
Functional classification of differentially expressed proteins in hemp under Pb stress. (**a**) Proteins up-regulated in BM; (**b**) proteins up-regulated in Y1.

**Figure 3 genes-10-00396-f003:**
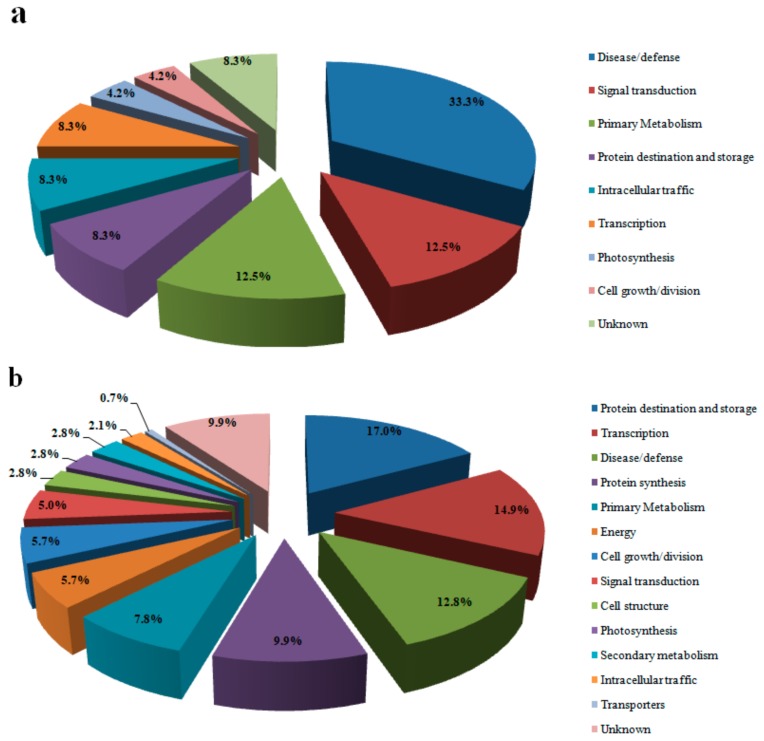
Functional classification of differentially expressed proteins in hemp under Pb stress. (**a**) Proteins down-regulated in BM; (**b**) proteins down-regulated in Y1.

**Table 1 genes-10-00396-t001:** Physiological indicators of two hemp varieties under Pb stress.

Treatment		Photo (μmolm^−2^·s^−1^)	Cond (mmol·m^−2^·s^−1^)	Ci (μl·L^−1^)	Tromol (g·m^−2^·h^−1^)	Chlorophyll (mg/g)	Soluble Sugar (mg/g)	SOD (U/g)	MDA (nmol/g)
BM	Pb	14.3 ± 0.46	0.51 ± 0.02	342.00 ± 6.00 ^*^	9.54 ± 0.29	68.31 ± 0.06	295.14 ± 0.65	193.12 ± 5.63	0.98 ± 0.38
	CK	15.0 ± 0.50	0.43 ± 0.09	310.33 ± 9.87	8.99 ± 0.32	68.75 ± 0.03 ^*^	304.13 ± 0.42 ^*^	195.00 ± 1.88	0.94 ± 0.17
	IR	-4.7%	17.6%	10.2%	6.1%	-0.6%	-3.0%	-1.0%	4.0%
Y1	Pb	14.03 ± 1.00	0.48 ± 0.02	322.67 ± 4.16 ^*^	8.24 ± 0.53	68.51 ± 0.11	288.47 ± 1.64	200.62 ± 3.75	0.86 ± 0.19 ^*^
	CK	17.33 ± 1.14 ^*^	0.55 ± 0.05	306.00 ± 2.65	9.74 ± 0.26 ^*^	68.92 ± 0.10 ^*^	292.94 ± 4.25	203.12 ± 9.44	0.53 ± 0.08
	IR	−19.0%	−12.8%	5.4%	−14.4%	−0.6%	−1.5%	−1.2%	62.3%

Note: The values represent mean ± SD of three replicates. * Significant difference at *p* < 0.05. IR = (Pb-CK)/CK × 100%. Photo, Net photosynthetic rate; Cond, stomatal conductance; Tromol, transpiration rate; CK, Control; IR: increment rate.

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
