# Peer review of "Protein Changes in Response to Lead Stress of Lead-Tolerant and Lead-Sensitive Industrial Hemp Using SWATH Technology"

_genes, 2019, doi:10.3390/genes10050396_

Round 1
Reviewer 1 Report
I have no more comment.
Reviewer 2 Report
The revisions to the paper are appropriate.
This manuscript is a resubmission of an earlier submission. The following is a list of the peer review reports and author responses from that submission.
Round 1
Reviewer 1 Report
p.p1 {margin: 0.0px 0.0px 0.0px 0.0px; font: 12.0px Helvetica} p.p2 {margin: 0.0px 0.0px 0.0px 0.0px; font: 12.0px Helvetica; min-height: 14.0px}1) The abstract should describe the plant tissues that were studied.
2) Leaves were chose for analysis. Given mechanisms of Pb resistance may be due to factors throughout the plant (roots, stem, leaves, etc), please provide a clear justification for this choice of tissue.
3) Most of the results section consists of description of the biology of proteins whose abundance was observed to change. No experimental validation of the changes in protein abundance, or of the proposed effects of these changes on plant biology, are performed.
4) Minimal attempts have been made to compare the changes in proteome observed in the two plant varieties. How similar where the Pb-dependent changes in proteomes in these varieties? Can this advise on the mechanisms of Pb resistance or Pb toxicity?
5) The MS data must be made available through a repository such as PRIDE.
6) Figure 1 should be moved to Results.
7) It is not clear exactly what samples were taken for proteomic analysis. The methods state that there were a total of 60 pots, with 6 hemp plants in each pot; 2-4 g of leaves were randomly collected; for a total of 12 samples (presumably Y1, Y1 Pb, BM, and BM Pb in triplicate). Please precisely define what samples were taken (e.g. were triplicate leaves taken from different plants in different pots?).
8) L106 “grinded” “ground”
9) L108 How long were samples incubated to allow them to be dissolved?
10) Were samples reduced / alkylated? No details are provided.
11) L112. Provide full details or precise reference for the FASP method used.
12) L117 “SWATH -labeled peptide mixture”. I can not find any mention of labelling details. Where peptides labelled? What was this mixture - were samples combined from different plants?
13) L125. Approximately how much material was injected for each analysis?
14) L149. Were P-value adjustments made for multiple testing?
15) L154. “coincide with rapid growth.” Please define this in more detail. Was this time point different for the different treatments? What were the time points?
16) Table 1. In the table, please define: Photo, Cond, Ci, Tromol, CK, IR.
Author Response
Response to Reviewer 1 Comments
Point 1: The abstract should describe the plant tissues that were studied.
Response 1: Revised as suggested. The plant tissue (leaves) has been added, please see the abstract, page 1 line 22.
Point 2: Leaves were chose for analysis. Given mechanisms of Pb resistance may be due to factors throughout the plant (roots, stem, leaves, etc), please provide a clear justification for this choice of tissue.
Response 2: The reasons for choosing leaves as the research object for analysing mechanisms of Pb resistance of hemp are as follows: Firstly, leaves are important nutrient organ of hemp, which accounts for a large proportion of the total biomass of hemp. Secondly, the leaf is the material and energy exchange centre of hemp, where photosynthesis and respiration are carried out. Thirdly, hemp leaves also absorb a large amount of lead, which also seriously affects their metabolic activities. In addition, we will continue research other organs in future research.
Point 3: Most of the results section consists of description of the biology of proteins whose abundance was observed to change. No experimental validation of the changes in protein abundance, or of the proposed effects of these changes on plant biology, are performed
Response 3: Thanks for your useful suggestions. In this paper, a large number of differential proteins with hemp response to Pb stress were obtained by proteomic analysis. These proteins are directly related to hemp adaptation to Pb stress. In the future study, we will use some technologies (such as gene clone, PRM etc.) to validate these important proteins.
Point 4: Minimal attempts have been made to compare the changes in proteome observed in the two plant varieties. How similar where the Pb-dependent changes in proteomes in these varieties? Can this advise on the mechanisms of Pb resistance or Pb toxicity?
Response 4: In this paper, A total of 63 and 372 proteins were recognized as differentially expressed proteins (≥1.5 fold) in BM and Y1, respectively, suggesting that the metabolic status and molecular physiological activities of the Pb tolerant variety BM were more stable than those of Pb susceptible variety under the stress, and the strong metabolic activity in Y1 resulted in the sensitivity to Pb stress. After further analysis, we found that the adaptability of the two varieties was significantly different under Pb stress. This result also reflects the difference in the mechanisms of Pb resistance or Pb toxicity for resistant variety (BM).
Point 5: The MS data must be made available through a repository such as PRIDE
Response 5: Revised as suggested. All MS data is uploaded to the iProX (https://www.iprox.org, IPX0001592000). We add this sentence in the page 4 line 141-142.
Point 6: Figure 1 should be moved to Results.
Response 6: Revised as suggested. The figure 1 has been moved to results. Please see page 5.
Point 7: It is not clear exactly what samples were taken for proteomic analysis. The methods state that there were a total of 60 pots, with 6 hemp plants in each pot; 2-4 g of leaves were randomly collected; for a total of 12 samples (presumably Y1, Y1 Pb, BM, and BM Pb in triplicate). Please precisely define what samples were taken (e.g. were triplicate leaves taken from different plants in different pots?).
Response 7: Revised as suggested. We have been modified this sentence. Please see page 3 line 96-98.
Point 8: L106 “grinded” “ground”.
Response 8: Revised as suggested. The error has been corrected. Please see page 3 line 101.
Point 9: L108 How long were samples incubated to allow them to be dissolved?
Response 9: The samples were dissolved with the lysis buffer by sonication for 10 mins. Please see page 3 line 103.
Point 10: Were samples reduced / alkylated? No details are provided.
Response 10: Yes,samples were reduced before digestion. Thank you. We have been added it, please see page 3 line 107-110.
Point 11: L112. Provide full details or precise reference for the FASP method used.
Response 11: Revised as suggested. We have been add the reference in the paper. Please see page 3 line 111.
Point 12: L117 “SWATH -labeled peptide mixture”. I can not find any mention of labelling details. Where peptides labelled? What was this mixture - were samples combined from different plants?
Response 12: Sorry, no peptides were labelled, and 10 μg peptides each sample were pooled for building library. Please see page 3 line 118, and page 3 line 111-112.
Point 13: L125. Approximately how much material was injected for each analysis?
Response 13: 2 μg of peptide was injected for each analysis. Please see page 3 line 127.
Point 14: L149. Were P-value adjustments made for multiple testing?
Response 14: No, because the sample size was small and the identified proteins were not enough for P-value adjustments. This standard can be found in the reference:
Belin S, Nawabi H, Wang C, et al. Injury-induced decline of intrinsic regenerative ability revealed by quantitative proteomics[J]. Neuron, 2015, 86(4): 1000-1014.
Point 15: L154. “coincide with rapid growth.” Please define this in more detail. Was this time point different for the different treatments? What were the time points?
Response 15: Revised as suggested. We have been modified it. Please see page 4 line 156-157.
Point 16 :Table 1. In the table, please define: Photo, Cond, Ci, Tromol, CK, IR.
Response 16: Revised as suggested. We have been added the abbreviate explanations in the table foot-note. Please see page 6.

Reviewer 2 Report
In this study, the authors used SWATH (sequential window acquisition of all theoretical mass spectra) technology to study the difference in proteomics between the lead(Pb)-tolerant seed-type hemp variety Bamahuoma (BM) and the Pb-sensitive fiber-type hemp variety Yunma 1(Y1) under Pb stress (3g/kg soil). they found that hemp adapted to Pb stress through accelerating ATP metabolism; enhancing respiration, light absorption and light energy transfer; promoting assimilation of intercellular nitrogen (N) and carbon (C); eliminating reactive oxygen species; regulating stomatal development and closure; improving exchange of water and CO2 in leaves; promoting intercellular transport; preventing aggregation of unfolded proteins; degrading misfolded proteins; and increasing the transmembrane transport of ATP in chloroplasts.
Major comments:
1. Page 3, line 91: about 40 days(?) cultivation, is it sufficient for hemp growth? their proteins or plant physiological functions were limited in the early 40 days lifecycle.
2. Were any Pb concentrations measured in these 4 groups? No direct evidence that Pb affected these proteomics. The study only compared the hemp with Pb soil to the hemp without Pb soil. The results were indirect, and mechanisms were hypothetic only. Please write a section to discuss the limitation of this study.
Minor comments:
Table 1 : too many abbreviates to realize. Please give abbreviate explanations in the table foot-note.
Author Response
Response to Reviewer 2 Comments
Point 1: Page 3, line 91: about 40 days(?) cultivation, is it sufficient for hemp growth? their proteins or plant physiological functions were limited in the early 40 days lifecycle.
Response 1: In the field, the hemp only takes 4-5 months in the whole growth period. After 40 days of growth, hemp can grow to more than 1250px in the field and 1000px in the pot. From this time on, hemp enters into the rapid growth period, and it can increase 1-125px a day. Also this period is the most active period of the proteins and metabolic activity of hemp throughout the whole growth period. Therefore, the analysis time point of this experiment is 40 days after sowing.
Point 2: Were any Pb concentrations measured in these 4 groups? No direct evidence that Pb affected these proteomics. The study only compared the hemp with Pb soil to the hemp without Pb soil. The results were indirect, and mechanisms were hypothetic only. Please write a section to discuss the limitation of this study.
Response 2: The difference between BM and Y1 response Pb stress can be well reflected by 8 physiological indicators measured in this paper (table 1), so we did not measured the Pb concentrations in different organs. Our research is also the same as the following studies, with no Pb concentrations measured (Li et al., 2016; Zhu et al., 2016; Dai et al., 2017) :
Dai, C., Cui, W.T., Pan, J.C., Xie, Y.J., Wang, J. and Shen, W.B., 2017. Proteomic analysis provides insights into the molecular bases of hydrogen gas-induced cadmium resistance in Medicago sativa. J Proteomics, 152: 109-120
Li, G.K., Gao, J., Peng, H., Shen, Y.O., Ding, H.P., Zhang, Z.M., Pan, G.T. and Lin, H.J., 2016. Proteomic changes in maize as a response to heavy metal (lead) stress revealed by iTRAQ quantitative proteomics. Genet Mol Res, 15:
Zhu, F.Y., Chan, W.L., Chen, M.X., Kong, R.P.W., Cai, C.X., Wang, Q.M., Zhang, J.H. and Lo, C., 2016. SWATH-MS Quantitative Proteomic Investigation Reveals a Role of Jasmonic Acid during Lead Response in Arabidopsis. J Proteome Res, 15: 3528-3539
A section to discuss the limitation of this study has been added in the section of conclusions. Please see page 14 line 478-484.
Point 3: too many abbreviates to realize. Please give abbreviate explanations in the table foot-note.
Response 3: Revised as suggested. We have been added the abbreviate explanations in the table foot-note. Please see page 6.

Round 2
Reviewer 1 Report
Key queries were not adequately addressed:
Point 3: Most of the results section consists of description of the biology of proteins whose abundance was observed to change. No experimental validation of the changes in protein abundance, or of the proposed effects of these changes on plant biology, are performed
Point 14: L149. Were P-value adjustments made for multiple testing?
Point 10: Were samples reduced / alkylated? No details are provided.
Response 10: Yes,samples were reduced before digestion. Thank you. We have been added it, please see page 3 line 107-110.
If samples are fractionated, failure to alkylate peptides may result in substantially suboptimal results.
Reviewer 2 Report
I have no more comments.